# Exploring the Role of Communication Asset Mapping (CAM) as a Strategy to Promote Hereditary Cancer Risk Assessment Information Within African American Communities

**DOI:** 10.3390/ijerph22010075

**Published:** 2025-01-08

**Authors:** Crystal Y. Lumpkins, Kimberly A. Kaphingst, Lynn R. Miller, Evelyn Cooper, Margaret Smith, Katie Belshe, Garry Lumpkins, Jill Peltzer, Prajakta Adsul, Ricardo Wray

**Affiliations:** 1Department of Communication, Huntsman Cancer Institute, University of Utah, Salt Lake City, UT 84112, USA; kim.kaphingst@hci.utah.edu; 2Faith Works Connecting for a Healthy Community, Kansas City, KS 66103, USA; lmllr3619@gmail.com (L.R.M.); cooperevelyn4242@gmail.com (E.C.); glumpkins@gmail.com (G.L.); 3Department of Family and Community Health, University of Kansas Medical Center, Kansas City, KS 66160, USA; msmith33@kumc.edu; 4Department of Cancer Navigation and Intake, The University of Kansas Health System Westwood, Westwood, KS 66205, USA; knelson15@kumc.edu; 5School of Nursing, University of Kansas Medical Center, Kansas City, KS 66160, USA; jpeltzer2@kumc.edu; 6Department of Internal Medicine, University of New Mexico, Albuquerque, NM 87131, USA; padsul@salud.unm.edu; 7Department of Behavioral Science and Health Equity, College for Public Health and Social Justice, Saint Louis University, Saint Louis, MO 63103, USA; ricardo.wray@slu.edu

**Keywords:** Communication Asset Mapping, cancer, hereditary risk assessment, African American

## Abstract

**Objective**: African Americans (AAs) carry the largest burden for almost every type of cancer in the US and are also more likely to die from cancer. Approximately 10% of cancers can be explained by a hereditary factor and detected earlier. Many AAs, however, have inequitable access to hereditary cancer risk assessment (HCRA) tools and information, further exacerbating disparities in cancer rates. Innovative communication strategies to promote community-based HCRA information have promise as a means encouraging optimal primary cancer screening among AAs. The current pilot study followed a participatory process where researchers engaged with a Community Advisory Board (CAB) to explore how Communication Asset Mapping (CAM) could assist lay health advisors with the dissemination of evidence-based HC/RA information within AA faith communities. **Methods**: The research team and CAB conducted exploratory community-engaged group discussions with residents (n = 21) guided by Communication Infrastructure Theory, and used a community-engaged mapping process to inform the development of a CAM dissemination strategy. **Results**: Through textual analysis, the following conclusions were reached: (1) optimal locations (e.g., community centers) within specified neighborhood networks should have representatives who are trusted ambassadors to assist with HCRA information dissemination; (2) trusted community member voices should fully represent the neighborhood network in the community-engagement mapping process; (3) well-known and frequented geographic locations should provide a true representation of participants’ neighborhoods to create a robust health information network concerning HCRA. **Conclusions**: Community residents appreciated the engagement process; however, they felt that its impact was limited due to the lack of community voices within their neighborhoods to identify important communication resources within the network for optimal HCRA information dissemination. CAM, therefore, is an important public health strategy for the identification of trusted networks and useful communication resources within these networks. The strategy was also helpful in pinpointing people who could be critical communicators of emerging health information akin to HCRA.

## 1. Introduction

The cancer health inequities and disparities among African American men and women are well documented [1,2,3]. Strategies to address cancer disparities among this group and other racial/ethnic minorities are, therefore, a top priority in national, regional, and local cancer prevention programs [4,5,6]. These efforts include the dissemination and implementation of evidence-based practices within community settings [7]. Community-based public health programs aiming to reach minoritized populations have incorporated factors to ensure cultural appropriateness; however, a gap remains in the identification of inequitable communication factors that may further exacerbate the optimal dissemination and implementation of evidence-based practices (EBP) [8,9,10,11]. The cancer burden among African Americans [12,13,14] and the information inequalities necessitate culturally appropriate strategies to mitigate both the disparities in primary and secondary cancer screenings and the information gaps to increase equitable outcomes [15,16,17,18]. In addressing these screening deficiencies, health promotion programs have evolved from individual-level [19] to multi-level interventions aiming to identify impediments to or facilitators of equitable cancer outcomes [20]. These factors account, in part, for an individual’s perceptions of well-being and their ability to respond in a given environment [21]. Integrating evidence-based best clinical practices and tools that address individual-level cancer risk within community settings may help reduce information inequity and other barriers among minoritized populations.

Hereditary cancer risk assessments (HCRA) [22,23,24,25,26] are a foundational service where precision medicine tools detect inheritable genetic variants that increase an individual’s risk for developing certain cancers, and are part of the national clinical EBP recommendations for those who have been diagnosed with specific cancers [22]. While these clinical tools are EBPs that are utilized by genetic counselors and other medical personnel, some groups, including under-represented and minoritized populations, are not aware of and do not receive these services or technologies [15,27,28,29,30,31,32]. Among African Americans, studies show that the uptake of HCRA tools and genetic testing services is significantly lower due to barriers that include limited awareness, a lack of recognition of risk, a lack of provider recommendation, and inadequate access [17,33]. African American populations are also less likely to undergo genetic testing once they are diagnosed with cancer, as well as being less likely to undergo preventive testing (prior to diagnosis) [34]. In one study, African American women were shown to have a lack of support in obtaining counseling and subsequent genetic testing in resource-limited settings [35]. This was, in part, a result of their having incomplete family health history information, an important factor in risk assessments for genetic counseling and testing recommendations.

Although HCRA and genetic testing services have positively impacted cancer outcomes, cancer risk assessments primarily benefited non-racial/ethnic minority communities [17,33,36]. Employing public health efforts that reach under-represented populations in community settings where existing public health promotion programming exists increases the likelihood of the uptake of such services [37]. Community- and participatory-focused strategies were shown to be effective in addressing the multi-level factors that impact cancer outcome across the cancer care continuum [20,38]. These strategies span from addressing individual-level cancer risk perceptions to addressing interpersonal, organizational [39], and systemic barriers. Concomitantly, cancer communication and information strategies, developed as part of the public health promotion programming and strategically targeted to minoritized populations, impact health risk perceptions, behavior, and cancer outcomes [40,41]. Perceptions about cancer risk, judgment of the acceptability of this risk, self-efficacy, trust in medical personnel, communication, and how information is exchanged within a defined environment are important factors affecting the target behavior (HCRA uptake). Here, too, communication about cancer risk may be impaired without targeted and culturally tailored information [9,42,43,44,45]. Engagement with racial/ethnic minority communities to explore attitudes, perceptions, and beliefs about HCRA [9,46] shows that these communities are receptive to different types of cancer prevention information. Further, integrating trusted community members into the HCRA information dissemination process has promise to help mitigate long-standing mistrust, fear, lack of access, and other barriers.

### CAM and CIT in HCRA Information Dissemination

Communication asset mapping (CAM) is a method recommended for the identification of communication asset resources within specified areas or neighborhoods [47,48,49,50,51].

CAM, also akin to the public health method ‘asset mapping’, is a theory-based fieldwork and participatory methodology [47] that allows for the mapping of “local geography and has an important role in shaping well-being” (p. 775).

The Communication Infrastructure Theory [52,53], often applied when conceptualizing the CAM process, is [49,52] a framework that describes the flow of communication and components and includes (1) a storytelling network component that comprises the creators and consumers of communication and information and (2) a communication action context that includes communication asset resources in the surrounding environment that may “facilitate or impede communication between creators and consumers” [49,54]. CIT increases the understanding of resources in the public information environment as “an ecological system wherein creators and consumers of communication at multiple levels, including individuals, social networks, community organizations, and media, are nested within a community where communication is shaped by geography, the built environment, and organizational and political structures and systems” [47] (p. 774). CIT provides a framework that helps determine which organizations or people within a community help shape the information that is received and how the information is spread from individuals to groups. In the context of HCRA information dissemination, the storytelling network (the best communication resources within a defined geographic area of African American communities) was explored; this was a communication network comprising community members, trusted organizations, and lay health advisors affiliated with these trusted organizations. This storytelling network was posited as an avenue for HCRA public health information dissemination [47,49] and a robust infrastructure for HCRA information.

With strong public health efforts and public health communication strategies aimed to help reduce cancer disparities in community settings [37,40,43,55], the barriers to disseminating evidence-based technologies and practices relating to the HCRA disparities within minoritized populations may be reduced. We used a participatory approach to identify the opportunities for HCRA information dissemination among African Americans. CIT was also used as part of the framework to engage African American residents in community discussion groups to identify potential assets in the communication resources. A pilot Communication Resource Asset Map was subsequently created as a tool to assist in disseminating HCRA information within African American faith communities in the Midwest. To the authors’ knowledge, this is one of the first studies to explore a community-engaged approach to HCRA information dissemination among an under-represented population.

## 2. Materials and Methods

### 2.1. Design

The research team used a participatory approach with an HCRA-focused study Community Advisory Board (CAB) and subsequently engaged with African American residents through community-engaged mapping (see Figure 1). Community-engaged mapping, a part of the CAM process, is described as “a group mapping exercise designed to answer specific research questions and gather feedback from community members for the purpose of developing place-based planning, policy and interventions. It can be described as a focus group around a map”, p. 10. The group discussions emphasized (1) a participatory and consensus-based process of ranking communication ‘asset’ resources for HCRA health (communication) in geographically defined neighborhoods and (2) capturing residents’ lived experiences within these geographically defined neighborhoods to inform a future CAM strategy intervention for HCRA health communication within densely populated African American areas. The geographically defined areas included two predominately African American Faith-Based Organizations (FBOs) and the surrounding neighborhoods, which served as the “storytelling” network for the HCRA. During community group discussions, research staff and CAB facilitators followed an a priori structured discussion guide to lead community members in ranking pre-selected communication resource assets (Appendix A). The script was co-developed based on a participatory asset mapping project [51] and the existing literature, and in partnership with the study’s CAB (a primary care physician, a physical therapist, a genetic counselor, a registered nurse, a religious leader, two IT professionals, and a cancer survivor). The facilitator guide included six sections: (1) Overview; (2) Mapping Introduction; (3) Review of the Asset Mapping Process and Example; (4) Small Group Discussion; (5) Large Group Discussion and Polling; (6) Wrap Up. In Section four, the CAB and research staff facilitated small group discussions simultaneously in virtual break-out rooms. Embedded into the small group discussions were questions about the pre-selected Communication Asset Resources for HCRA health communication. Facilitators asked the following questions:

What: Of the places on the map, which places are the best communication assets? Are there any missing?

Why: Why do you consider these places communication assets?

What about the location: Is there something about where they’re located that makes it a communication asset?

Why: Why don’t you consider these places communication assets?

What can be improved? Of all the places on the map, are there places on the map that could become communication assets or could be improved as communication assets? What kind of improvements?

A virtual format for group discussions was adopted to address the limitations to meeting in person that occurred during the pandemic and in accordance with the participants’ preference.

### 2.2. Recruitment of Participants, Ethical Considerations, Data Collection and Study Participants

#### 2.2.1. Recruitment of Participants

The research team and IRB-trained CAB members recruited community resident participants through purposive and snowball sampling [56] between August and September 2021. CAB members were also asked to assist with the recruitment of residents and individuals within their own community networks who either lived in or were familiar with the geographically defined areas of the community-engaged group discussions. The neighborhoods surrounding the study FBOs served as the primary recruitment target area; secondary recruitment was carried out using snowball sampling. Participants who were eligible identified as African American; were 18 or older; were familiar with the residential areas near the study churches in Kansas and Missouri; and could identify African American businesses, organizations and establishments for cancer prevention as communication resources for HCRA health information dissemination.

#### 2.2.2. Ethical Considerations

Prior to participation in the study, community resident participants received a copy of the consent form via email. Participants later received a secure link via email and consented to the study prior to taking a demographic survey. After completion of the survey, participants were contacted by research staff or an IRB-trained CAB member with a calendar invitation to participate in the community group discussions. On the day of the discussions, study staff asked if participants had any additional questions. The study was approved by the Institutional Review Board at the University of Kansas Medical Center (#00147613).

#### 2.2.3. Data Collection

The CAB (Faith Works Connecting for a Healthy Community) engaged with the research study team from October 2020 to September 2021 to prepare for data collection with residents and individuals familiar with the neighborhoods surrounding two FBOs (one in Kansas and one in Missouri). On the day of the community-engaged group discussions, the research team and CAB met 30 min prior to the event and pre-assigned participants to a virtual breakout discussion group. At the start of the workshop, participants were let into the main room, received a welcome from the CAB chair and Principal Investigator, and read a script about their participation. After the introduction, instructions, and review of the consent form, the participants were asked if they had any questions and were subsequently sent to their pre-assigned break out room. Two CAB members who also worked in the Information Technology (IT) industry created scoring sheets (PDFs) where participants could rank the communication resources within the defined geographic areas for the two FBOs. The IT CAB members also created two Google maps, using the CAB and research staff’s pre-selection process to identify geographic locations within a two-mile radius of the FBOs (see Appendix B).

Facilitators were asked to record their session for data collection and data analysis and to participate in a debriefing session after the closing segment. Participants were thanked for their time and given instructions on how to receive an incentives (USD 50 electronic Wal-Mart gift cards). Participants were also informed that the research team would follow up with them to administer a post-survey and share the results within the next six months.

#### 2.2.4. Study Participants

Participants (n = 21) were primarily female (90%), highly educated (postbaccalaureate/graduate degrees), and 60 years of age. More than half the sample (57%) had a household income of USD 50,000 to USD 100,000 a year. The majority had a primary care provider (PCP) and saw the provider once a year (52%) or several times a year (43%) but had not talked with their PCP about HCRA (86%). In the post-survey, the number of participants who had talked to the PCP about HCRA had risen slightly (9%), from three to five participants (Table 1).

### 2.3. Data Analysis

The Principal Investigator met with two IRB-trained CAB members to conduct a qualitative coding and thematic analysis. CAB members went through two 2 h qualitative coding trainings before individually coding five community group discussion transcripts. Individual coding led to a constant comparison analysis of the five group discussions that were conducted with the aforementioned a priori structured interview guide (What, Why, What about the Location, What Can Be Improved?). Coders met three times to perform a constant comparison coding [57] of the overlapping themes among all group discussions. Once the individual coding was completed, coders met and noted the emerging themes across all group discussions.

## 3. Results

The conclusions that emerged from the group discussions included the following: (1) optimal locations within specified neighborhood networks should have representatives who are trusted ambassadors to assist with HCRA information dissemination; (2) trusted community member voices should fully represent the neighborhood network in the community-engagement mapping process; and (3) well-known and frequented geographic locations should provide a true representation of participants’ neighborhoods to create a robust health information network concerning HCRA.

Theme 1: Optimal locations (e.g., community centers) within specified neighborhood networks should have representatives who are trusted ambassadors to assist with HCRA information dissemination

The first theme shows that those who participants perceived as important people and locations that were perceived as important should be included as assets in the communication resources for HCRA information dissemination. The pre-identified locations were natural extensions of the places within residents’ networks that contain an individual who could serve as a trusted health ambassador for HCRA information dissemination. It was important that these communication asset resources were relatable (age), accessible (time), and respected (notoriety) within the defined geographic area. The representative responses among participants reflect specific locations and people within their networks who could assist in optimal HCRA information dissemination.

Participant (Group 3): “You have the children there (Boys and Girls Club) but you have their parents, their grandparents or someone picking them up and the information could be there. So, when we’re thinking about prevention, it’s never too early to provide education, it’s never too early to start talking about prevention because if we think across the large or the wider spectrum, we always talk about how we want to teach our children prevention in terms of don’t use drugs, don’t drink alcohol and they start at around seven or eight. If we’re saying which would be a good place if we want to talk about prevention, we have to start early and so the Boys and Girls Club have all of those people coming in and out of there at any given time”.

Participant (Group 5): “When you look at, I think it’s Number 3 (communication asset resource ranking), which churches worship could be valuable to us, it’s a time sensitive situation. As we all know, October is Breast Cancer Month and if we could get something done to those churches doing Breast Cancer Month, I think that will be very impact (full). At Church X, we actually have a breast cancer ministry. Well, it’s not breast. It’s a cancer ministry and so they deal with not only breast cancer. They deal with any cancer the third Sunday is the emphasis Sunday for that event. If you could work around being time sensitive with it, the churches would be a great avenue to really plug into”.

Theme 2: Trusted community member voices should fully represent the neighborhood network in the community-engagement mapping process

A second theme emerged among the sample, emphasizing who was absent from the community-mapping process. Participants expressed that community members were missing from the group discussions and that this would impact the overall process of developing an inclusive CAM strategy. Participant comments from the group discussions emphasized the importance of representation from organizations within the neighborhoods that focused on youth and specific media outlets in the area that placed emphasis on health and families.

Participant (Group 2): “There’s something else I thought about and I don’t know how you would feel about this but what about the black newspaper?”.

Participant (Group 3): “And the other thing is that again with the different generations because there are multigenerational people coming in and out of there (location) and so we have to look at and think about those people and the people that are coming in and out and we look at their view of their family. Many of them have experienced some type of illness, major illness, traumatic illness within their families and so a lot of them have more information than we know”.

Theme 3: Well-known and frequented geographic locations should provide a true representation of participants’ neighborhoods to create a robust health information network concerning HCRA.

The CAB and research staff attempted to include an exhaustive list of communication asset resources prior to the group discussions. The pre-planned poll included the communication asset resources that CAB and researchers had ranked and deemed important to incorporate into group discussions and ranking process. Participants, however, expressed that some important resources were missing from the provided ranking system (ABCD or asset-based community development) [51].

Participant (Group 2): “Don’t leave out the schools because the schools have parent groups that are still trying to be active and be pertinent and present things for families and [crosstalk] families of school children could be going through some of these issues and so the information could be right there at the school”.

Participant (Group 3): “I think that Brush Creek Community Center is a good location and I think it’s a good place to disseminate information”.

Participant (Group 4) “And something that I’m thinking out of the box. We’re looking at brick and mortar places but what about public transportation like buses and you can put flyers and stuff on the buses?”.

## 4. Discussion

Conducting community-engaged mapping group discussions with a dedicated study CAB was demonstrated to be a feasible and robust participatory approach to conduct a qualitative inquiry [51]. The process contributed to the development of a CAM strategy for HCRA information dissemination. Here, the participation of trusted community leaders and HCRA experts led to a rich discussion of a little-known and difficult subject [58,59]. To conduct virtual community-engaged mapping group discussions, it was important to have a diverse and dedicated group (CAB and research team) to recruit, prepare for, and conduct the discussions. As with any community-engaged research project, it was important to maintain research ethics and include stakeholders in all phases of the research process [60,61].

The group modified an “in-person” process for community engagement-mapping [48] and adopted this for online group discussions during and after the 2020 pandemic. Although the discussions led to the development of a CAM strategy, there were key considerations during this process that should be noted. The development of this type of strategy via a virtual platform emphasized the importance among the CAB of including an assessment of digital literacy and an assessment of asset mapping knowledge prior to the group discussions. The findings from this exploratory study provide detailed methodological guidance for conducting community-engaged mapping discussions in medically underserved and minoritized communities. In this study, we focused on (1) exploring the use of community-engaged mapping discussions to develop a CAM strategy (e.g., a communication asset resource map) that can serve as a dissemination platform to improve awareness, knowledge and (2) the prospects of providing further HCRA information/implementing an HCRA program within African American populations. The creation of the foundation for an information-rich infrastructure may provide a robust environment for HCRA information dissemination and subsequent program implementation.

Using the community-engaged discussions, participants were able to collectively rank and map the communication resource assets within two geographically defined FBO networks in the Midwest to develop a CAM strategy. The resulting pilot communication resource asset map is suggested to be an instrumental strategy in identifying and assisting trusted lay health advisors and other influential communicators within the storytelling network with the dissemination of HCRA information. These voices may also be influential in building an information infrastructure regarding HCRA within their own community organizations.

The development of a tailored CAM strategy for improving HCRA information dissemination among minoritized populations holds promise in addressing cancer disparities and cancer health outcomes. In the present project, the engagement of community members throughout the process of developing a pilot CAM strategy helped to identify community asset-oriented communication resources within specific geographic locations, where HCRA information dissemination could address the deficit in awareness of HCRA. This process also allowed the research team to identify places in which lay health advisors in African American communities may bolster HCRA information and education. Because FBOs have served as viable places for evidence-based [62] cancer interventions and health communication programs, which are effective at reaching African American communities [55,63], it is plausible that these organizations are also acceptable and feasible community settings in which HCRA information can be disseminated and health education programs may be widely implemented. This study explored, based on previous research on African American faith community members’ positive perceptions of HCRA tools and technology [46], the role that CAM strategies may play in guiding HCRA information dissemination and the future implementation of HCRA tools and technologies.

Although HCRA tools (e.g., family health histories) have been clinically available for the last 30 years, a significant portion of the general population is still unaware that these tools and technologies exist [64]. Further, patient-facing tools within communities are nearly non-existent and/or are not feasible to implement. As an example, in a recent national survey about genetic counseling and testing, oncologists reported that they were less likely to refer African American women than European American women [65]. Our exploratory studies show the value of these programs among these populations. African American faith and other minoritized communities are receptive to this type of information [9,46] and to provider recommendations of these technologies. Clinical efforts, in tandem with ongoing community-based efforts, provide multiple avenues by which communities may benefit from cancer prevention tools and reduce cancer disparities. The use of CAM within communities could inform clearly defined dissemination strategies within the clinical setting as well. This strategy, as part of culturally tailored dissemination and implementation (D & I) strategies, has the potential to also aid in clinical decisions, increase the integration of genetic literacy tools among patients, and shape lay communication in clinical settings. While clinical environments and settings differ in size and operation compared to community-based settings, there are considerations that require assessment and evaluation in both settings (e.g., credibility, trust, timing, reach, and effectiveness). The CAM approach is a context-specific dissemination and implementation strategy that considers cultural contexts and also aims to increase information equity within a uniquely defined information infrastructure or communication action context. This research provides an insight into where community awareness efforts can be targeted and optimized for at-risk individuals to provide them with the motivation to undergo an HCRA.

The focus of the project was to identify the targets of change within an engagement communication action context for HCRA information dissemination. Here, the goal was to leverage existing African American FBO partnerships, an HCRA-focused CAB, and previously collected psycho-social data concerning HCRA risk assessment and known barriers and facilitators among African Americans concerning cancer risk and prevention [66,67] to inform our participatory research process. In addition, the CIT provided a guide to identify specific communication resources within a defined geographic area and network as integral to HCRA information dissemination. The findings from the group discussions show that trusted messengers within faith community networks with lived experience were important representatives that could act as credible creators and disseminators of HCRA information. While community members were not queried about specific HCRA messaging and content, it is worth noting that message elements are critical to how disseminators could shape this information to increase the dissemination of specific types of HCRA information, as well as increasing awareness and knowledge outcomes. Depending on what and how risk messages (type of HCRA, dosage and framing of risk probabilities as a loss or win, etc.) are incorporated into the HCRA information, the messaging may serve as a barrier to or facilitator of the adoption of CAM as a dissemination strategy. Community-engaged approaches that facilitate the co-creation and development of messaging strategies increase their relevance and acceptability, as community voices are embedded into the process. The distinction here is the role of CAM itself in improving the dissemination, and subsequently the adoption, of the strategy within a specific information environment or setting. The role of CAM and leveraging communication asset resources is critical to promote EBPs within African American faith networks. The findings from this study further support the role of CAM in informing the implementation strategies regarding public health.

## 5. Limitations

The reception to the community group discussions was overall positive; however, there were some limitations that the weakened the results of the process and the consideration of CAM as a strategy for HCRA information dissemination. While the FBOs were located in a bi-state area, individual participants in these organizations were only familiar with activities within their state of residency (Kansas or Missouri), thus limiting their ability to rank the communication assets within the target areas. This sample was also highly educated and motivated, which could have lessened their identification of assets that may be more beneficial for those with fewer financial or educational resources. Participants also had varying levels of computer literacy skills but were willing to hop online because someone they knew was present in the discussion. Dissimilar skills among participants extended the time spent on group discussions and also their comprehension at the conclusion of the session and in the post-survey. Although some community members were noted as being absent from the group discussions, the CAB chair led efforts to recruit many of the key community members who were needed to identify important communication resources.

## 6. Study Strengths

This study adds to the health communication and D &I science literature by explicating the methodology around CAM as a strategy for HCRA information dissemination within African American communities. Following additional inquiry, CAM may also be applicable in the implementation of EPBs within clinical settings for minoritized individuals. The process of identifying communication asset resources and leveraging the networks within the neighborhoods and communities in which these populations live and discuss cancer and other health prevention information may inform how, when, where and what type of HCRA is culturally appropriate to adopt or adapt and eventually implement or de-implement. Another attempt to increase the centrality of health communication and information in the D & I process of HCRA EBPs may be found in Albright and colleagues’ [68] efforts. Albright et al. discuss the critical role of communication in facilitating the implementation of evidence-based care (psychotherapy) in community mental health settings. Their study findings emphasized the need to include delineating and identifying components and methods for effective communication when communication is included in the “exploration and preparation phases of implementation” of an EBP (p. 324). The authors also state that out of the 73 identified implementation strategies, “the field of implementation science has largely left unexcavated the details of how communication may be utilized to facilitate implementation, thus leaving it (communication) hiding in plain sight” (p. 325). Expanding how information around health is communicated, specifically precision medicine, is warranted given the disparities in cancer communication and the absence of precision medicine information for minoritized populations. More research is needed to explore and test the effectiveness of communication strategies, including CAM, which is designed to bolster the dissemination and subsequent adoption and implementation of EBPs. Dissemination research, which is an overlapping but understudied field [69] compared to health communication research, is an important focus for the field of D & I science. The findings from this study have important theoretical implications for the role of communication in disseminating EBPs, which could draw from the beginnings of Implementation Science and the pioneering work of Everett Rogers’ Diffusion of Innovations [70].

## 7. Conclusions

The inclusion of a focus on D & I research and its synergy with health communication approaches show promise in informing strategies to increase the awareness of precision medicine and improve the adoption of strategies to disseminate and implement EPBs that improve cancer health outcomes. Study participants and CAB members reiterated the importance of having trusted messengers in specific places within the community who could serve as capable ambassadors, disseminating HCRA information and other pertinent information through trusted networks. The empirical testing of risk-framed HCRA messages at multiple levels (individual, interpersonal, organizational, and community), and discovering which specific components impact behavior, could be the subjects of future public health communication studies. The use of different types of communication strategies to disseminate evidence-based HCRA practice could also be a topic for future study. These areas of research also have practical and research implications for the use of D & I in similar precision medicine programs and in other minoritized communities and settings, as well as to reduce health inequities and cancer mortality for all.

## Figures and Tables

**Figure 1 ijerph-22-00075-f001:**
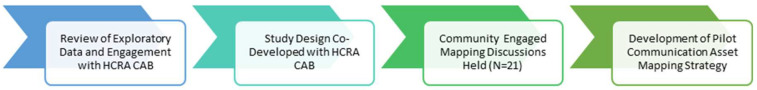
Community engagement, and the design and development of the pilot CAM.

**Table 1 ijerph-22-00075-t001:** Socio-demographic information (n = 21).

Age Gender Men (n = 2) Women (n = 19) Income * Residency Kansas (n = 4) Missouri (n = 17) Education Medical Provider Frequency of visits to the PCP Talked to a PCP about HCRA Prior to Community Group Discussion (n = 3) Talked to a PCP about HCRA Post Community Group Discussion (n = 2)	Age Mean ($\bar{x}$) 60 10% 90% 57% (USD 50–100 K); 19% 81% Post-Baccalaureate/Graduate Education (57%) 100% 52% Primary Care Provider (PCP) once a year 86% had not talked with their PCP about HCRA; 14% of the sample had talked with a PCP 9% increase

* Some residents who lived in Missouri or Kansas were affiliated with the study FBOs in the opposite state of their residency.

## Data Availability

Data supporting reported results can be requested by contacting the corresponding author.

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
