# Peer review of "Exploring the Role of Communication Asset Mapping (CAM) as a Strategy to Promote Hereditary Cancer Risk Assessment Information Within African American Communities"

_ijerph, 2025, doi:10.3390/ijerph22010075_

Round 1
Reviewer 1 Report
Comments and Suggestions for Authors
The following are concerns and comments needed to be addressed to consider this manuscript for publication:
1. Please include more references for the first sentence in the introduction.
2. Please include sub-headings in the methods section to define your sample size, recruitment strategy, data collection, and plan for data analysis, along with ethical statement/IRB approval.
3. For the type of qualitative analysis carried out, was it thematic qualitative analysis? Which qualitative framework was followed to integrate CAM into analysis?
4. There is a typo in the word "online" in line 304. Please correct.
5. Please spell out "2" in line 317.
6. The conclusion mentions D&I; however, this was not addressed prior. Please make sure that D&I is mentioned in the introduction/methods prior to the discussion.
7. Please make sure to add a subheading for your limitations section.
Author Response
We thank the reviewer for his/her/their comments and have responded to each comment below:
The following are concerns and comments needed to be addressed to consider this manuscript for publication:
1. Please include more references for the first sentence in the introduction.
Response: We have included more references to the first sentence in the introduction.
2. Please include sub-headings in the methods section to define your sample size, recruitment strategy, data collection, and plan for data analysis, along with ethical statement/IRB approval.
We have significantly reorganized the methods section and have included sub-headings.
3. For the type of qualitative analysis carried out, was it thematic qualitative analysis? Which qualitative framework was followed to integrate CAM into analysis?
The authors thank the reviewers for this question. We have detailed the type of qualitative analysis carried out. This work is a community-engaged mapping method or CEM similar to focus groups guided by Communication Infrastructure Theory. We have defined and described this within the Methods section of the manuscript.
4. There is a typo in the word "online" in line 304. Please correct.
The authors thank the reviewer for this catch and have corrected this typo.
5. Please spell out "2" in line 317.
The authors have corrected this.
6. The conclusion mentions D&I; however, this was not addressed prior. Please make sure that D&I is mentioned in the introduction/methods prior to the discussion.
The authors have re-organized the introdcution and methods to address D & I in these sections of the manuscript.
7. Please make sure to add a subheading for your limitations section.
The authors have added a sub-heading for the limitations section.
Reviewer 2 Report
Comments and Suggestions for Authors
This manuscript proposed communication asset mapping (CAM) for hereditary cancer risk assessment in African American. The sample size of participants is too small (n=19) for derive statistically significant findings. Also, the results section only covered some statistics and themes from CAM discussion. It’s still unknown how many benefits this strategy can bring to African American in cancer prevent and personalized treatment. More quantitative assessment of the proposed strategy will be necessary.
Author Response
Reviewer Comment: This manuscript proposed communication asset mapping (CAM) for hereditary cancer risk assessment in African American. The sample size of participants is too small (n=19) for derive statistically significant findings. Also, the results section only covered some statistics and themes from CAM discussion. It’s still unknown how many benefits this strategy can bring to African American in cancer prevent and personalized treatment. More quantitative assessment of the proposed strategy will be necessary.
Author Response: The authors thank the reviewer for the comment and why it appears to be a non-representative sample. The study is a pilot project and was to be the first step in gathering formative data for a future pragmatic study. The sample was not to be generalizable but informative to gain more information about the concept and feasibililty of Communication Asset Mapping. This step was to gather information collaboratively with a CAB/community partners from African American communities/genetic counselors, etc. The primary aim of the project was not to measure engagement through a deductive process but rather to query individuals prior to the community discussions to gather some demographic information that would be helpful with describing the participants in the sample. The reviewer is correct to state the following: "It’s still unknown how many benefits this strategy can bring to African American in cancer prevent and personalized treatment. More quantitative assessment of the proposed strategy will be necessary." The next phase for this project will be to take information from this pilot study to then test the feasibility of the strategy. The authors have taken steps to make sure that this is clear in the abstract and also in the introduction and conclusion of the manuscript.
Reviewer 3 Report
Comments and Suggestions for Authors
Intro= adequate
Methods - The methods includes references and a detailed description of CAM which should be moved to the introduction. Only methods - what they did - not why they did it - should be in the methods section.
It says "Following IRB approval #00147613, through the University of Kansas Medical Center, recruitment began August 2021 and concluded September 2021." Then later it says "The CAB engaged with the research study team and 2 Faith-Based Organizations (FBOs) from October 2020 to September 2021 to discuss results from exploratory cancer-related genetic counseling and testing projects that would inform the present project." Can the authors say more about the CAB? who were these people? why was the CAB created? Is it a standing CAB? Who supports the CAB? Your cancer center or was it established for this work specifically? Do the differences in dates explain the length of time for preparation? Was this project co-developed with the CAB?
Purposive sampling was used to recruit the participants. What was the criteria for inclusion? How were these people selected? 26 were identified but only 21 are in the table. What happened to the other 5? Vaguely the paper describes that the scientific staff and CAB led the CAM. How many CAB members? again, so little is included about the CAB. Were members of the CAB members of the FBOs?
Why were the FBOs selected? Why only a 2 mile radius? Why was this process done online and not in-person given the community nature of this work? Can you describe the community that these FBOs are in? It seems that most of the study participants are highly educated - s
Might the authors include details about how the analysis was done? It seems that only included description of coding training but left out the after methods for the analysis.
Also, the methods focuses on qualitative analysis and the CAM but doesn't really share details or an analysis plan for the surveys. The Results section mentions a pre/post survey. How many took each survey? What kinds of questions were asked in them?
Was there IRB oversight for this work? Please describe the consenting process.
Results: Did only 5 people take the post survey?
Why are quotes repeated from table into text? Why is there no text accompanying the quotes to provide context? This results seems like a quote dump that doesn't really provide readers and understanding of the value. Might there be a list prepared of where the participants said the work should be done? Is that really what this is about? You all had discussion that identified places and possible people to leverage to offer HCRA to AAs from these 2 well-off churches? You say you used CAM methodology...but how you described this just seems like an online discussion about places to offer health education in the community.
Discussion - The discussion highlights the online aspect as being new. The intro and the early text in the methods barely mentions this "novel" aspect and yet so much time is spent on this in the discussion. It wasn't even clear why it needed to be online. What was the advantage? Much of the lengthy text of the discussion is more background that an explanation of the results. Is this really considered Dissemination & Implementation (D &I) Science? Conclusions are often shorter summaries yet this one goes on and on and includes even more citations.
There are so many good parts of this paper - the intro and discussion sections could be parts of a great review paper on the topic. The descriptions of the work itself is lacking. The study findings are vague and seem insignificant despite this reviewer being excited about their potential.
70 references is a lot.
Author Response
The reviewer's comments and review are greatly appreciated. The authors respond to the reviewer's points below:
Reviewer's Comment:
Methods - The methods includes references and a detailed description of CAM which should be moved to the introduction. Only methods - what they did - not why they did it - should be in the methods section.
It says "Following IRB approval #00147613, through the University of Kansas Medical Center, recruitment began August 2021 and concluded September 2021." Then later it says "The CAB engaged with the research study team and 2 Faith-Based Organizations (FBOs) from October 2020 to September 2021 to discuss results from exploratory cancer-related genetic counseling and testing projects that would inform the present project." Can the authors say more about the CAB? who were these people? why was the CAB created? Is it a standing CAB? Who supports the CAB? Your cancer center or was it established for this work specifically? Do the differences in dates explain the length of time for preparation? Was this project co-developed with the CAB?
Authors' response: The reviewer has several questions that were addressed and included in the re-organized Methods section. The CAB is a standing HCRA-focused CAB that is affiliated with the Cancer Center. The differences in dates do explain the length of time for preparation and the project was co-developed by the CAB. These details are outlined in the revised version.
Reviewer's Comment:
Purposive sampling was used to recruit the participants. What was the criteria for inclusion? How were these people selected? 26 were identified but only 21 are in the table. What happened to the other 5? Vaguely the paper describes that the scientific staff and CAB led the CAM. How many CAB members? again, so little is included about the CAB. Were members of the CAB members of the FBOs?
Authors' Response: The authors thank the reviewer for these questions. The revised version is rewritten to clarify where this could be confusing. There were 5 CAB members and 21 participants. The submitted version included CAB members as participants and this was not the case. The other questions about the CAB are also detailed in the revised version.
Reviewer's Comment:
Why were the FBOs selected? Why only a 2 mile radius? Why was this process done online and not in-person given the community nature of this work? Can you describe the community that these FBOs are in? It seems that most of the study participants are highly educated - s
Authors' Response: The rationale for why they were selected and why the process was done online are also addressed in the revised version (the ongoing impact of the 2020 Pandemic; the study was conducted in October, 2021). In addition, we describe the community that the FBOs are in which are African American faith communities. Individuals were affiliated with the selected FBOs. The rationale for the geographic or areas are also outlined based on literature and a community research engagement project that is cited in the revised manuscript.
Reviewer's Comment:
Might the authors include details about how the analysis was done? It seems that only included description of coding training but left out the after methods for the analysis.
Authors' response: These details about how the analysis was done is added. The authors have also added a graphic to show the overall process and also included the facilitator guide as supplementary materials.
Reviewer's Comment:
Also, the methods focuses on qualitative analysis and the CAM but doesn't really share details or an analysis plan for the surveys. The Results section mentions a pre/post survey. How many took each survey? What kinds of questions were asked in them?
Authors' Response: The survey was done to gather specific demographic information (age, income, insurance coverage, state, etc.) to: 1. Prepare for the group discussions and 2. capture HCRA (one item) for a future study. All participants took the survey. The results are included in a table at the end of the manuscript. The survey was brief and was not to measure or test any hypotheses; this was an inductive study. The authors wanted to capture additional data that would help juxtapose to qualitative data and information. The authors are more than willing to include the brief survey for readers as supplementary material.
Reviewer's Comment:
Was there IRB oversight for this work? Please describe the consenting process.
Authors' response: Yes, there was IRB approval. This was included in the first version, however, it was included in the Ethical Consideration within the Methods section in this version. The consenting process is also described.
Reviewer's Comment:
Results: Did only 5 people take the post survey?
Author's Response: 21 people took the post survey. The number of participants that we refer to in this statement (5) were the participants who had completed HCRA.
Reviewer's Comment:
Why are quotes repeated from table into text? Why is there no text accompanying the quotes to provide context? This results seems like a quote dump that doesn't really provide readers and understanding of the value. Might there be a list prepared of where the participants said the work should be done? Is that really what this is about? You all had discussion that identified places and possible people to leverage to offer HCRA to AAs from these 2 well-off churches? You say you used CAM methodology...but how you described this just seems like an online discussion about places to offer health education in the community.
Authors' response: The table with quotes have been removed and we have provided context within the revised version. These appear to be quote dumps but are actually taken from structured group discussions/ facilitator guide where the facilitator is asking participants to primarily rank communication resources. Therefore, the quotes that are included may not be traditional or conversational as are focus groups discussions. We have provided a facilitator guide to show how these discussions were facilitated and why it may appear to be a quote dump. The authors have also attempted to clarify the differences between the method and strategy of CAM. CAM is both a method and strategy where community members were engaged in the mapping of communication resources and thus creating a map for strategically disseminating information about HCRA. The engagement, this project and effort are also more than just an online discussion but a process that began in 2018 and this is just one culmunation of that and other projects that were ongoing during this time period. In addition, the project was to be held in person but because of the pandemic, there was a need to adapt.
Reviewer's Comment:
Discussion - The discussion highlights the online aspect as being new. The intro and the early text in the methods barely mentions this "novel" aspect and yet so much time is spent on this in the discussion. It wasn't even clear why it needed to be online. What was the advantage? Much of the lengthy text of the discussion is more background that an explanation of the results. Is this really considered Dissemination & Implementation (D &I) Science? Conclusions are often shorter summaries yet this one goes on and on and includes even more citations.
There are so many good parts of this paper - the intro and discussion sections could be parts of a great review paper on the topic. The descriptions of the work itself is lacking. The study findings are vague and seem insignificant despite this reviewer being excited about their potential.
70 references is a lot.
Authors comments: We thank the reviewer for his/her/their comment. We have reorganized the paper to pull these elements together in a more succint fashion. We do believe that the project is novel and that the amount of references are warranted because there is an integration of scholarship and also making a case for this research. In fact, another reviewer asked for additional references in the introduction.
We re-wrote the manuscript to highlight the significance of this work and that the engagement with these populations, interdisciplinary work between scholars and also the fact that CAB members led some of the research speaks to why community-engaged health communication may be bolstered by these efforts.
Round 2
Reviewer 1 Report
Comments and Suggestions for Authors
We thank the authors for addressing all suggested changes. No additional comments from my end.
Author Response
Comments 1: We thank the authors for addressing all suggested changes. No additional comments from my end.
Authors Response 1: We the authors thank the reviewer for these comments
Reviewer 3 Report
Comments and Suggestions for Authors
Changes to the Intro and Methods sections were generally responsive to the review. There are still some issues with the Results and Discussion sections.
Suggestion - Make 3 sections that each describe one theme. Before including the quoted statements, say something like " Representative responses of this theme include:..." Currently, the text no longer discusses themes 2 and 3 and again the participants' responses are dropped in without any lead in or transition statements.
Discussion - the conclusion is huge. Conclusions in this reviewer's view should be relatively brief highlighting the impact of the work. The current conclusion includes more discussion so some of this should be moved up into the discussion. Some of the text are study strengths which could easily follow or even preceed the limitations.
Acronymns are inconsistently used. For example, sometimes the CAM is spelled out, other times it is an acronym. Might the authors spell it out at the start and reliably use the acronym thereafter?
Author Response
Comments 1.
Changes to the Intro and Methods sections were generally responsive to the review. There are still some issues with the Results and Discussion sections.
Authors' Response: The authors thank the reviewer(s) for the comments and have revised the Intro and Methods sections as well as re-organizing the results and discussion sections.
Comment 2.
Suggestion - Make 3 sections that each describe one theme. Before including the quoted statements, say something like " Representative responses of this theme include:..." Currently, the text no longer discusses themes 2 and 3 and again the participants' responses are dropped in without any lead in or transition statements.
Authors' Response: The authors thank the reviewer(s) for the comments and have re-organized this section and added text to introduce or lead into the participant comments/quotes.
Comment 3.
Discussion - the conclusion is huge. Conclusions in this reviewer's view should be relatively brief highlighting the impact of the work. The current conclusion includes more discussion so some of this should be moved up into the discussion. Some of the text are study strengths which could easily follow or even preceed the limitations.
Authors' Response: The authors thank the reviewer(s) for the comments and have re-organized the discussion section by editing, taking out copy and re-organizing this section.
Comment 4.
Acronymns are inconsistently used. For example, sometimes the CAM is spelled out, other times it is an acronym. Might the authors spell it out at the start and reliably use the acronym thereafter?
Authors' response: The authors thank the reviewer(s) for the comments and have gone through the entire manuscript to add consistency for CAM and other acroynms also.